# Patients with Positive Lymph Nodes after Radical Prostatectomy and Pelvic Lymphadenectomy—Do We Know the Proper Way of Management?

**DOI:** 10.3390/cancers14092326

**Published:** 2022-05-08

**Authors:** Bartosz Małkiewicz, Miłosz Knura, Małgorzata Łątkowska, Maximilian Kobylański, Krystian Nagi, Dawid Janczak, Joanna Chorbińska, Wojciech Krajewski, Jakub Karwacki, Tomasz Szydełko

**Affiliations:** 1Department of Minimally Invasive and Robotic Urology, University Center of Excellence in Urology, Wroclaw Medical University, 50-566 Wroclaw, Poland; gosialatkowska@gmail.com (M.Ł.); maxkobylanski@gmail.com (M.K.); krystian.nagi@student.umw.edu.pl (K.N.); dawid.janczak@umw.edu.pl (D.J.); joanna.chorbinska@student.umw.edu.pl (J.C.); wojciech.krajewski@umw.edu.pl (W.K.); tomasz.szydelko@umw.edu.pl (T.S.); 2Department of Biochemistry, Faculty of Medical Sciences in Katowice, Medical University of Silesia, 40-752 Katowice, Poland; knura.milosz@gmail.com

**Keywords:** prostate cancer, lymph node invasion, radical prostatectomy

## Abstract

**Simple Summary:**

Prostate cancer (PCa) is the second most frequent malignancy in the male population worldwide. Men with a nodal invasion established after radical prostatectomy with lymph node dissection are a heterogeneous group of patients who require more thorough stratification and therapy individualization, which remain uncovered by current guidelines. Considering a multitude of prognostic factors and novel diagnostic techniques, classifying patients into narrower and more specified risk groups should be a vital part of lymph node positive PCa management in the future.

**Abstract:**

Lymph node invasion in prostate cancer is a significant prognostic factor indicating worse prognosis. While it significantly affects both survival rates and recurrence, proper management remains a controversial and unsolved issue. The thorough evaluation of risk factors associated with nodal involvement, such as lymph node density or extracapsular extension, is crucial to establish the potential expansion of the disease and to substratify patients clinically. There are multiple strategies that may be employed for patients with positive lymph nodes. Nowadays, therapeutic methods are generally based on observation, radiotherapy, and androgen deprivation therapy. However, the current guidelines are incoherent in terms of the most effective management approach. Future management strategies are expected to make use of novel diagnostic tools and therapies, such as photodynamic therapy or diagnostic imaging with prostate-specific membrane antigen. Nevertheless, this heterogeneous group of men remains a great therapeutic concern, and both the clarification of the guidelines and the optimal substratification of patients are required.

## 1. Introduction

Prostate cancer (PCa) is the second most common cancer in the male population worldwide, with an estimated 1,400,000 new cases annually. It is the fifth-leading cause of cancer-related death among men, with a total of over 375,000 deaths each year [1].

The prognosis among PCa patients varies and depends on many individual disease characteristics. Accurate staging is necessary for appropriate risk assessment and choosing further therapeutic measures. Multiple staging procedures, including digital rectal examination (DRE), prostate-specific antigen (PSA) serum levels, transrectal ultrasonography (TRUS), positron emission tomography–computed tomography (PET/CT), and multiparametric magnetic resonance imaging (mpMRI), are aimed at evaluating PCa expanse preoperatively. Nevertheless, pelvic lymph node dissection (PLND) and postoperative evaluation by a pathologist is still the most sensitive procedure for detecting lymph node invasion (LNI) and as such remains the gold standard in PCa staging. Lymph node metastases (LNM) are present among 3–80% of PCa patients after radical prostatectomy (RP) with extended pelvic lymph node dissection (ePLND) [2].

The occurrence of LNM is a relevant prognostic factor, as it affects cancer-specific survival (CSS), as well as recurrence, and thus its proper detection and assessment are crucial in the choice of the post-prostatectomy therapeutic pathway [3]. Most therapeutic strategies are essentially based on observation with or without salvage therapy, external beam radiation therapy (EBRT), and/or androgen deprivation therapy (ADT). However, both American and European guidelines concerning the management of patients with nodal invasion after RP with ePLND provide no clear path of the disease’s further diagnostic and therapeutic processes.

The purpose of this narrative review is to sum up the available data on prognostic factors and treatment options in patients with pathological LNM detected after RP (pN+) in order to confront them with therapeutic strategies proposed in the available guidelines and to search for optimal clinical solutions.

## 2. Risk and Prognostic Factors for Nodal Involvement

There are many known risk factors associated with the presence of pelvic lymph node metastases and worse prognosis in pN+ PCa patients. These include classic factors as well as pathological features of the excised tissues (Figure 1).

### 2.1. PSA

The predictive role of PSA in pN+ patients is of particular importance, as it has been commonly used in clinical practice. Postoperative PSA takes part in the therapeutic procedure; more precisely, it serves as an indicator of the effectiveness of RP and of biochemical recurrence. It is a feature that allows for the monitoring of the possible progression of the disease after prostatectomy. During the observation of pN+ patients, as soon as PSA increases, more aggressive management should be introduced. Depending on the clinical center, the PSA level that indicates the threshold for biochemical recurrence varies between 0.1 and 0.4 ng/mL [4]. Preoperative PSA is an important value, as it is a known prognostic factor for pN+ patients. Swanson et al. noted that, among patients with PSA below 10, 25% of them had biochemical failure in comparison with 57% among those with PSA equal to or higher than 10 (*p* < 0.0001). Importantly, this correlation was observed regardless of other disease characteristics [5]. However, preoperative PSA has not always been recognized as an independent predictor. In a study that demonstrated an association between preoperative PSA and the advent of biochemical progression, the median time to BCR for patients with a PSA level greater than 20 ng/mL was significantly shorter (16 months) than for those with a PSA level of 10.1 to 20 ng/mL (27 months) or others with lower PSA levels (*p* = 0.0005). However, the difference in the results among the groups of patients with low PSA (2.6 to 4 ng/mL versus < 2.6 ng/mL) was not statistically significant (*p* = 0.4). The preoperative PSA correlated with the interval to biochemical cancer progression. However, a multivariate analysis, which included other characteristics, such as Gleason score or tumor stage, revealed that PSA should not be considered a significant independent prognostic factor [6].

### 2.2. T—Tumor Stage

Available nomograms aimed at predicting nodal involvement consist of different clinicopathological features. The “size and extent” of the tumor, reflected in the T stage of the American Joint Committee on Cancer (AJCC) TNM staging system, is often one of them. It is present among the most utilized predictive tools, including Briganti, Memorial Sloan Kettering Cancer Center (MSKCC), and Partin nomograms. Multiple studies have proven that T stage is an independent risk factor for nodal metastases, where a higher T stage corresponds with a higher risk of LNI [7,8,9]. The results of the systematic review of population-based evidence clearly show that the risk of LNI in men with pT2 disease is significantly lower compared to more advanced disease [10]. In relation to therapeutic decisions, Zaremba et al. analyzed a cohort of 7791 pN+ patients from The National Cancer Database. They found that men with higher stage tumors were more likely to receive any treatment than undergo observation and were more likely to be managed with combination therapy than ADT alone. Additionally, this study confirmed that locally advanced (pT3b-T4) disease was found to be an independent predictor of worse OS in pN+ patients [11].

### 2.3. ECE—Extracapsular Extension

The extracapsular extension (ECE) refers to local tumor growth beyond the prostate pseudocapsule and corresponds to the pT3a stage. The presence of ECE in patients after RP + ePLND is associated with a higher incidence of LNI. Di Trapani, in a study on 742 patients treated with robot-assisted radical prostatectomy (RARP) + ePLND, tested novel ECE score on the basis of MRI findings. The authors found that pN+ patients more frequently showed ECE 5 lesions (63.7 vs. 20.5%; *P* < 0.001) in comparison with pN− patients. Similarly, in a multivariable logistic regression model, a higher ECE score corresponded with higher rates of LNI (ECE 4 vs. 3 OR: 2.99; ECE 5 vs. 3 OR: 6.97; *p* < 0.001). The evaluation of extracapsular extensions is important in planning the treatment of patients undergoing RP. Currently, preoperative imaging diagnostics such as multiparametric magnetic resonance are used for this purpose, which helps to predict the risk of ECE prior to surgery. To calculate the probability of LNI in PCa patients, Trapani et al. developed a novel nomogram including an mpMRI-derived ECE score. The model based on the fixed covariables PSA, ISUP group, and percentage of positive cores with the addition of the ECE variable turned out to be the most accurate for calculating the patient probability of LNI. This significant association between a high mpMRI-derived ECE score and the subsequent presence of LNI could be used in the treatment process to more precisely identify candidates for ePLND and to increase the therapeutic potential of this procedure [12].

### 2.4. GS—Gleason SCORE

The degree of histological differentiation constitutes one of the most important prognostic factors. A higher Gleason score correlates with worse clinical results. Wei et al. studied the use of artificial intelligence in assessing the impact of risk factors on the occurrence of nodal metastases. Using machine learning, they developed a predictive model to assist clinicians in decision making when performing PLND. Among the clinicopathological features, they identified PSA level, T stage of disease, percentage of positive cores, tumor size, and Gleason score to be independent risk factors. In a multivariate logistic regression model, statistically significant ORs of 2.414 (1.315–4.431, *p* = 0.004), 4.393 (2.321–8.315, *p* < 0.001), and 5.696 (2.986–10.867, *p* < 0.001) were reached for Gleason scores of 7, 8, and ≥9, respectively. Moreover, in a performed extreme gradient boosting model calculating the relative importance of clinical features on the presence of LNI, Gleason score was proven to be the most significant among the abovementioned factors [9].

### 2.5. GGG—Gleason Grade Group

Apart from the value of the Gleason score itself, an important prognostic aspect is the affiliation with the relevant Gleason Grade Group (GGG). The differentiated oncologic outcome between Gleason score 7 patients was highlighted by the International Society of Urological Pathology (ISUP) in 2014 with the introduction of Grade Group 2 for GS 3 + 4 and Grade Group 3 for GS 4 + 3. Currently, ISUP distinguishes five different Grade Groups in its classification. Porcaro et al. analyzed clinical factors associated with the presence of LNI in a group of 184 patients staged preoperatively as cN0. Of these, in 33 patients the histopathological examination revealed nodal metastases. Patients who had LNI showed a higher percentage of biopsy Gleason pattern >4 + 3. In a multivariate regression model one of the most impactful risk factors was biopsy Gleason pattern >4 + 3, which corresponds with GGG > 3 (OR 3.666; *p* = 0.004) [13]. Another study was conducted on 316 cT2N0M0 patients who underwent radical prostatectomy, 46 of which turned out to be pN+ after surgery. Multivariate logistic regression analysis showed that GGG was a risk factor for LNI in all patients (*p* = 0.030, OR 1.517 (95% CI, 1.042–2.209)), and that GGG ≥ 3 in particular corresponded with a higher risk of nodal metastases (*p* = 0.029, OR 1.767 (95% CI, 1.061–2.943)) [14].

### 2.6. Cribriform Pattern

Among the histopathological factors that influence the presence of lymph node metastases in PCa patients is the cribriform architecture. Due to the architectural heterogeneity of Gleason Grade 4, ISUP introduced four growth patterns in 2014, including ill-formed, fused, glomeruloid, and cribriform. Studies have proven the cribriform pattern to be an adverse predictor for the presence of LNI. In a study on patients with a Gleason score of 8, cribriform growth was more common among the pN+ patients (19% versus 3%, *p* = 0.05) [15]. Likewise, in a study on patients with a Gleason score of 7, cribriform was an unfavorable factor as its presence was more often associated with lymph node metastases (11% vs. 2.6% in the absence of cribriform pattern, *p* = 0.037) [16]. On the contrary, cribriform pattern has no prognostic significance when located in the lymph nodes [17].

### 2.7. IDC—Intraductal Carcinoma

Another histopathological characteristic influencing prognosis is the presence of intraductal carcinoma (IDC). It is believed that atypical cancer cells may spread via existing acini and ducts. This ductal spread leads to the formation of a type of architecture that is morphologically similar to a cribriform pattern, but in the case of IDC, the preservation of basal cells is maintained. The presence of IDC seems to be an adverse factor related to the development of LNI. Kimura et al. reported that positive nodal metastases were more common among patients with IDC of the prostate than in non-IDC cases (24% versus 4.9%) [18]. In a study on pN+ patients and lymph node metastasis’ correlation with histopathological patterns, the incidence of IDC was significant (identified in 65% of prostates). However, there are not many studies linking the occurrence of IDC with pN+ patients’ prognosis. Due to the more frequent presence of this feature among patients with metastases to the lymph nodes, further research on this correlation should be conducted [17].

### 2.8. pN—Pathological Node Stage

Various studies have established that a major risk factor associated with increased death from PCa is constituted by the number of lymph nodes containing micrometastases. Up to this moment, there is no widely accepted consensus on the exact number of positive nodes unambiguously translating into higher cancer-specific mortality. Touijer et al. found the presence of three or more positive nodes to be associated with a higher risk of cancer-specific death [4]. Boorjian et al., on the other hand, assessed having two or more positive modes as an independent predictor of mortality [19]. A cut-off value for cancer-specific survival of more than two positive lymph nodes was established by Briganti et al., indicating a worse CSS in those patients compared to men with up to two positive nodes [20]. Qin et al. conducted a cohort study on a Chinese population, also finding an increased risk of biochemical recurrence from prostate cancer after RP occurs in patients with two or more positive nodes [21]. Similarly, Stolzenbach et al. established having three or more positive lymph nodes as an independent risk factor of BCR [22].

### 2.9. LND—Lymph Node Density

Lymph node density is the ratio of the number of positive lymph nodes removed to the overall number of lymph nodes removed. This variable is closely related to a well-established prognostic factor—number of pN+. Froehner et al., in their study on the prognostic factors of pN+ patients, distinguished low lymph node density as an independent predictor of a good prognosis. Patients with above or equal to the median of 11.1% lymph node density were associated with a higher prostate cancer mortality (hazard ratio (HR) 1.66, 95% CI 1.04–2.64, *p* = 0.0340). On the contrary, a higher total number of removed lymph nodes did not correlate with a better survival rate [23]. An approach to evaluating oncological outcome based on positive lymph node density was also made by Daneshmand et al., showing increased risk for clinical recurrence in individuals with a >20% ratio of positive nodes to the complete number of nodes removed during surgery [24]. Likewise, a study on pN+ patients’ prognosis also confirms the independent prognostic role of lymph node density. Lower lymph node density was associated with a better CSS (an average density 11.1%). In addition, a 30% cut-off was presented as a key value. Exceeding this threshold might be a suggestion for the introduction of adjuvant systemic therapy as it significantly aggravates patient prognosis [25].

### 2.10. LVI—Lymphovascular Invasion

LVI is defined as the unequivocal presence of tumor cells within an endothelium-lined space. The presence of LVI is associated with a risk of LNI. LVI has been associated with a worse prognosis in patients after RP in many studies; however, not always as an independent adverse factor. Additionally, Jiang’s meta-analysis focused on the association of LVI with the clinicopathological features of PCa, such as LNM expansion. It was noted that patients with LVI were at a higher risk of having LNM (OR = 18.56, 95% CI: 7.82–44.06, *p* < 0.001) [26,27,28,29].

Moreover, in last year’s study presenting a CNN-based algorithm that identified patients at risk of LNM, LVI was proven to be an independent predictor for LNM (OR 11.73, 95% CI 3.96–35.70; *p* < 0.001). Currently, LVI is not considered a risk factor among predictive models for lymph node status in PCa patients, as it is rarely diagnosed in prostate biopsies. Consequently, it has been suggested that LVI be included in predictive nomograms and that its presence could be an indicator of ePLND [30].

### 2.11. ENE—Extranodal Extension

Extranodal extension (ENE) is defined as the presence of cancer cells outside the LN capsule infiltrating into the peri-nodal tissue. Its presence is usually associated with more aggressive disease characteristics such as higher cancer stage and grade and more positive lymph nodes [31]. The role of ENE as a predictor of oncological outcome in prostate cancer remains controversial. Downes et al. retrospectively analyzed the histopathological characteristics of nodal metastases on patient outcomes. On a group of 110 pN+ patients, they found ENE not to be prognostic for any oncological outcome [17]. Conversely, Luchini et al. performed a meta-analysis involving 1113 pN+ PCa patients, of which 658 were ENE(+). Their findings connect the presence of ENE to a higher risk of experiencing BCR compared to pN+ ENE(−) patients, but at the same time, no significance with regard to all-cause mortality or cancer-specific mortality was proven [32].

### 2.12. mpMRI

At present, imaging diagnostics are being increasingly applied in the risk stratification of LNI prior to radical prostatectomy. Nevertheless, current guidelines focus mainly on the clinical staging parameters. Among the most commonly used predictive tools for the assessment of LNI are the updated Briganti nomogram and the MSKCC nomogram, both of which lack information from preoperative imaging procedures. Novel predictive nomograms developed on the basis of a combination of clinical and radiological parameters may allow for a more precise risk stratification of patients who are candidates for ePLND. Multiparametric magnetic resonance imaging (mpMRI) gained importance some time ago as its introduction improved the accuracy of preoperative risk stratification. A study conducted by Porpiglia et al. was aimed at evaluating the role of mpMRI in assessing whether or not to perform PLND in patients with a LNI below 5% according to the Briganti nomogram. In a group of patients with an estimated preoperative risk of LNI below 5%, who underwent RP with PLND on the basis of the parameters maintained in mpMRI, the percentage of pN+ patients turned out to be similar when compared to the cohort with preoperative LNI risk higher than 5% (15% vs. 19.2%, respectively). In a multivariate analysis, three mpMRI parameters were proven to be significant independent predictors of LNI—extracapsular extension, seminal vesicle invasion, and predominant Gleason pattern 4 (OR 2.82 (95% CI 1.30–6.14), 3.44 (95% CI 1.76–6.72), and 2.30 (95% CI 1.09–5.28), respectively). Therefore, data obtained in mpMRI could be of use in assessing the risk of nodal metastases among patients with a low risk of LNI according to the Briganti nomogram [33]. In another study, Gandaglia et al. hypothesized about the applicability of old nomograms to contemporary patients undergoing mpMRI and targeted biopsy. On the basis of mpMRI data, including maximum diameter of index lesion, tumor volume, T stage (including presence of extracapsular extension and seminal vesicle invasion), and percentage of cores with clinically significant PCa, the authors created a nomogram that is helpful in prediction of LNI prior to surgery [34]. The need to define unified nodal imaging criteria led to the introduction of Node-RADS, a promising approach to the stratification of MRI-detected lymph nodes according to their potential metastatic status. On the basis of the imaging criteria of the “size” and “configuration” of the inspected lymph node, an assessment category is evaluated, ranging from “1—very low” to “5—very high” suspicion of LNI [35].

### 2.13. PSMA PET

Prostate-specific membrane antigen (PSMA), also known as N-acetyl-L-aspartyl-L-glutamate peptidase 1 (NAALAD1) or glutamate carboxypeptidase 2, is an enzyme expressed by most prostate cancer cells and, in combination with different molecules such as gallium-68 or fluorine-18, is used as a radiopharmaceutical in PET/MRI and PET/CT [36,37]. Although multiple studies point out the multiple advantages of PET/MRI over PET/CT, both are considered relatively effective tools in primary lesion detection and restaging [38]. Lately, there has been an ongoing discussion as to whether PSMA PET should affect further diagnostic and therapeutic approaches in patients requiring restaging, as it has been proven to be a promising diagnostic tool for nodal lesion detection [39,40]. In a narrative review by Kase et al., the authors proposed three possible scenarios and clinical implications of various PSMA PET/CT and conventional imaging result combinations. This study suggests that that some patients with negative PSMA PET/CT could benefit from salvage RT (SRT). Others, with positive LN on PSMA PET/CT and negative with conventional imaging, should be counseled on SRT- or ADT-based combinations [41]. Nevertheless, there is still no information on PSMA PET results in pN+ patients and the impact it would have on the management in this heterogeneous group. A decision-making process based on a PSMA PET examination may be the future solution for pN+ patients, but further retrospective and prospective studies are required.

### 2.14. Obesity—Higher Body Mass Index (BMI)

Tafuri et al., in their study conducted on 481 patients after RP with PLND, revealed that higher body mass index (BMI) enhances the risk of multiple LNI in comparison with patients without LNI (odds ratio (OR) = 1.147; *p* = 0.018) and in patients with a single positive node (OR = 1.189, *p* = 0.027) [42]. These results find support among patients who underwent robotic RP + ePLND; BMI independently increased the risk of multiple LNI in multivariate analysis (OR = 1.194, *p* = 0.026) [43]. In a retrospective study (2.997 PCa patients who underwent RP), obesity increased the risk of LNI compared with overweight and normal weight patients [44].

Considering the fact that the number of positive lymph nodes after RP with ePLND is associated with worse prognosis among pN1 PCa patients, recent studies indicate that obesity enhances the risk of number of LNI and thus worsens outcomes after RP with PLND.

### 2.15. Diabetes

The impact of diabetes on the risk of LNI is not clearly established. On the basis of the analysis of Anatomical Therapeutic Chemical (ATC) code level 2, Krönig et al. showed among 454 patients after RP with PLND that diabetes enhanced the risk of LNI (OR 2.869, *p* = 0.004) [45]. Lutz, in his study, demonstrates that, after RP with PLND, 10% (7/74) of diabetic patients presented LNI in comparison to only 1% (2/148) of the non-diabetic patients (*p* = 0.005) [46].

However, Kelkar et al. found a statistically insignificant association between diabetes and metastasis among both obese and nonobese diabetic patients after RP with PLND [47].

### 2.16. Testosterone

Androgens play a pivotal role in the development and pathogenesis of PCa [48]. In the work of Porcaro et al., the serum concentration of endogenous testosterone was inversely related to multiple pelvic LNI in comparison to no metastasis (OR 0.997; CI 0.994–1; *p* = 0.027) and single pelvic LNI (OR 0.994; 95% CI 0.989–1.000; *p* = 0.015) after RP with PLND [49]. Endogenous testosterone density, defined as the ratio of endogenous testosterone to prostate volume, demonstrated the same inverse relation with the prevalence of pelvic LNI among 201 patients with PCa classified as high-risk according to the European Association of Urology (EAU) [50]. Currently, there are not enough good-quality studies available to clearly predict the prognostic value of endogenous testosterone load.

## 3. Impact of PLND

Pelvic lymphadenectomy and the extent of its templates are currently under discussion, as the correct pattern remains an unsolved issue. The ePLND remains the gold standard procedure, both as a diagnostic instrument in nodal staging and as a therapeutic tool. The limited PLND (lPLND) covers the obturator fossa only, while the ePLND template (sometimes classified as standard PLND) additionally comprises external and internal iliac LNs, correctly staging 94% of patients [51,52]. The super-extended PLND consists of the abovementioned lymph nodes as well as internal and common iliac nodes and hypogastric and presacral LNs [53]. Although the consensus is reasonably established, it remains unclear as to which template is the most valuable for patients, as the lymph drainage pattern of the prostate is complex and oncological benefits seem to be very limited [54,55]. Nevertheless, the impact of the PLND template on pN+ potential management strategies is fairy negligible, as despite the extent of PLND use, the therapy remains diverse but singular. One should mention the risk of over-approach while pursuing broader and more refined PLND patterns. The Will Rogers phenomenon is a statistical paradox that may concern clinicians that originates from the superior diagnostic efficiency of more extensive templates [56]. There is a certain number of patients that would previously have been classified as pN0, while superextended lymphadenectomy would stage them as pN+. Thus, both new pN0 and pN+ populations would present more favorable outcomes due to the migration of this pN0–pN+ border group to the pN+ subpopulation. This potential statistical bias could affect future researchers’ conclusions and hinder interpretations of survival analyses [57]. In conclusion, the ePLND remains the gold standard nodal staging procedure, although its oncological advantage is being called into question and requires further investigation.

## 4. pN+ Management Options

### 4.1. Observation

Pathologically node-positive patients after radical prostatectomy represent a heterogeneous population. Depending on the disease characteristics, an appropriate method of further treatment can be chosen (Figure 2). Therefore, observation as a postoperative management strategy has its beneficial oncological outcomes only in a selected group of men. The term observation is defined as refraining from treatment until BCR occurs and then salvage therapies are introduced.

Not all node-positive patients are affected by a systemic disease. Touijer et al. reported that almost one-third of patients with LNI (28%) were free of BCR 10 years after RP and PLND alone. The estimated 5- and 10-year overall survival rates were 91% (95% confidence interval (CI), 86–94) and 60% (95% CI, 49–69), respectively. The estimated 5- and 10-year cancer-specific survival was also favorable (94 and 72%). The study recommends a risk-stratified approach. Factors such as increased Gleason score (>7), increased PSA, and high nodal metastatic burden (three or more positive nodes removed) correlated with an increased risk of BCR after RP. The metastasis in seminal vesicle and higher Gleason score (>7) were associated with increased risk of death after RP [4]. Similarly, Gupta distinguished adverse pathological features such as ≥pT3b stage of PCa, Gleason score ≥ 9, at least three pathologically positive nodes, and positive surgical margins. Men who lacked any of these characteristics did not benefit from any adjuvant therapy. In these patients, initial observation may be a better management strategy to avoid complications caused by non-essential treatment [58]. 

In comparative studies, observation is generally associated with less favorable outcomes than adjuvant therapies. Tilki et al., in a study of 773 patients, found aRT to be more beneficial for pN+ patients than no treatment or salvage radiation therapy (sRT) at the time of BCR. The 4-year metastasis-free survival (MFS) was 91.8% vs. 82.5% for aRT and observation, respectively [59]. Another study, which analyzed different treatment strategies after RP, showed that observation led to worse survival outcomes than adjuvant therapies with one exception. Even though ADT had preferable CSS in comparison with observation (HR: 0.64, 95% CI: 0.43–0.95), it was also linked to an increased risk of other-cause mortality (HR: 3.05, 95% CI: 1.45–6.40), which resulted in similar overall survival (OS) for ADT and observation (HR: 0.90, 95% CI: 0.65–1.25, *p* = 0.5) [60]. Additionally, Gupta in his analysis claimed that there was no difference in OS between ADT and observation (HR 1.01, 95% CI 0.87−1.18, *p* = 0.88). Conversely, a trial by the Eastern Cooperative Oncology Group (ECOG) found that early ADT compared with deferred treatment was associated with better overall survival (HR: 1.84, 95% CI 1.01–3.35, *p* = 0.04) [61]. However, the study has been repeatedly criticized due to the lack of PSA level controls and withholding therapy until osseous metastasis developed.

The EAU guidelines recommend considering expectant management in pN+ patients after PLND when ≤2 nodes and PSA level < 0,1 ng/mL. Marra et al. compared many studies and observed the presence of prognostic factors such as undetectable postoperative PSA, maximum two positive nodes, negative surgical margins, and low pathological stage, where observation can be considered [62]. In cases of patients with less advanced features, observation until developing BCR may be a good alternative to avoid unnecessary exposure to treatment complications. This risk-stratified approach should guarantee both oncological safety and quality of life.

### 4.2. ADT

The use of pharmacological methods to reduce androgen levels to the castration level is of key importance in the treatment of patients with hormone-dependent. In the past, the dominant view was that the presence of nodal metastases denoted a disseminated disease requiring systemic treatment. This paradigm changed, as subsequent studies revealed excellent oncological outcomes in patients with nodal metastases in whom ADT was not administered. A study by Touijer et al. compared patient outcomes between different postoperative management strategies. The analysis showed that there was no significant difference in OS between patients treated with ADT and observation, although a lower rate of CSM was observed for the ADT group [60].

The only prospective randomized phase III clinical trial evaluating the impact of immediate vs. deferred ADT after RP on oncological results was conducted by Messing et al. in 1999, as part of the ECOG, on a cohort of 98 patients [63]. The results of this study indicated a beneficial impact on OS and CSS in patients in whom early postoperative ADT was initiated. Observation or deferred application of ADT was associated with a threefold higher risk of death compared to the immediate ADT group. However, the results obtained by Messing et al. are of limited application in the contemporary treatment of PCa pN+ patients. The group of patients studied in this trial consisted of men with a high-volume nodal burden, a high percentage of positive surgical margins, and other oncological characteristics indicating advanced disease, and therefore of patients clearly benefiting from androgen deprivation. Since the clinical introduction of common PSA screening, the likelihood of developing multiple nodal metastases decreased, and hence the modern PCa patient is unlikely to benefit to the same extent as the patients in the ECOG trial.

Apart from these findings, it should also be taken into consideration that there are multiple adverse effects associated with ADT [64,65,66,67]. This raises the question in whom to appropriately start ADT treatment. Studies show that patients with aggressive disease characteristics are usually the ones in whom this kind of treatment is introduced most frequently. A systematic review on pN+ PCa patients after RP with PLND by Marra et al. mentions six retrospective studies, including 1319 men treated with aADT. In 39% of the cases, the patients had three or more positive nodes [62]. Another attempt to answer this question was made by Touijer et al. in creating a point system based on tumor characteristics that quantifies the survival benefit of adjuvant ADT + EBRT compared to no adjuvant treatment. In accordance with this, a clinical decision as to whether to start ADT could be based upon this model [60].

The last decade has seen the emergence of second-generation NSAAs, including enzalutamide and apalutamide, introduced in 2012 and 2018, respectively, which are showing greater AR blocking and decreased side effects in comparison to first-generation drugs [68,69]. New NSAAs are still under development, but there are already promising results in the treatment of patients with PCa, as shown by the trials involving the use of apalutamide or enzalutamide [70,71]. In 2019, the results of a randomized trial involving another NSAA (darolutamide) were published [72]. Fizazi et al. reported a prolonged metastasis-free survival of 40.4 months after administration of darolutamide vs. 18.4 months in the placebo group (HR 0.41, 95% CI 0.34–0.50, *p* < 0.001), with a comparable adverse effect incidence rate in both groups.

In conclusion, as pointed out earlier, patients with beneficial oncologic status, e.g., a low-volume nodal burden and no distant metastases, can profit from a watchful waiting strategy. However, in the case of patients with a higher nodal burden, as well as other more aggressive oncological characteristics, the application of ADT seems to be justified, despite its potential side effects. Therefore, a precise selection of patients potentially benefiting from ADT in accordance with their clinical condition and pathological nodal status is of key importance.

### 4.3. RT and ADT

Adjuvant RT combined with ADT is another treatment strategy that should be taken into consideration in patients with node-positive PCa without distant metastasis. Current guidelines recommend multiple strategies, among which is the combination of RT and ADT or ADT alone [73,74,75,76]. RT commonly consists of radiation to the prostatic fossa, pelvic lymph nodes, or whole pelvis. RT alone is not recommended and is rarely performed in patients with these characteristics, although the improvement of OS in high-risk and locally advanced PCa associated with this technique is clear [77].

There is strong evidence that the addition of adjuvant RT to ADT is both statistically and clinically significant when compared to ADT alone in pN1 patients, while patients with which characteristics will benefit the most remains unclear [78].

Firstly, it is crucial to mention that most of the collected data are based on retrospective studies. Guo et al. analyzed five studies for their meta-analysis and concluded that the addition of RT to ADT was associated with statistically significant benefits in terms of both OS (HR: 0.74) and CSS (HR: 0.40) in comparison with ADT alone [79]. The authors of the study described the results as a dramatic improvement but pointed out the need for the cautious selection of patients. Touijer et al. observed that patients benefiting the most from the combined therapy are those with higher-risk disease [60]. Abdollah et al., on the other hand, reported that aRT with ADT improved CSS and OS only in intermediate or high-risk patients [80]. Gupta et al. indicated in their cohort study certain pathologic features that enabled the identification of the population that benefited from aRT + ADT: ≥pT3b pathologic stage, Gleason score ≥9, ≥3 positive nodes involved, or positive surgical margin status. Men with one or more adverse features gained survival benefit from the described strategy, while up to 30% of patients without these hallmarks did not confer an increase in OS [58]. In the recently published systematic review by Marra et al. investigating 26 studies (23 retrospective and 3 randomized clinical trials), it was again suggested that the choice of therapeutic strategy be based on the clinical and histopathological aspects, e.g., the Gleason score, quantity of positive nodes, burden of surgical margin, or pathological stage and aggressiveness. Hence, observation strategy could be used in patients with lower-risk features: undetectable postoperative PSA, <3 positive nodes, negative margins, and nonaggressive histology, while in patients with higher-risk histopathological characteristics, RT and/or ADT should be taken into consideration [62]. Last but not least, a recent study by Bravi et al. showed results that were not promising; no significant improvement in OS or CSS was found when comparing ADT alone with the combined therapy [81].

It is also worth noting that some data suggest that aRT with ADT in comparison with ADT alone is associated with improved OS in patients with nodal metastasis identified using staging tools different from histopathological discovery after RP (modern imaging techniques, cN1 patients) [82].

In conclusion, one of the future goals would be to specifically identify patients that will benefit the most from the combined RT + ADT procedure. Certain histopathologic features should be precisely indicated, and the definition of the most profiting patient ought to be established. This will hopefully reduce morbidity and complications associated with overtreating patients with combined therapy. Further studies are required since most of the data are based on retrospective studies.

### 4.4. Docetaxel and New Antiandrogens (Abiraterone Acetate, Enzalutamide, Apalutamide)

Although they are not mentioned in official guidelines, several interventional studies indirectly looking for new multimodal therapeutic options among patients with pathologically confirmed LMN after RP have been conducted. Chemotherapeutics (mainly docetaxel), new antiandrogens, and others (including potentially photodynamic therapy) have been considered. Docetaxel is a semi-synthetic derivative of paclitaxel and belongs to the chemotherapeutic group of taxane. Its mechanism of action is caused by its binding to beta tubulin (stabilizing microtubules), which consequently leads to apoptosis and cell cycle arrest [83]. Docetaxel remains the first-line therapy in metastatic PCa resistant to castration [84]. The toxicity of chemohormonal toxicity seems to be tolerable. In a population of 42 patients with a median follow-up of 3.4 years, the outcomes of adjuvant high-dose intensity-modulated RT with docetaxel and long-term ADT brought about favorable effects regarding toxicity and clinical results, and the authors suggest this multimodal treatment for patients with LNM after RP. The trial included 16 (38.1%) patients with LNM, but the results of this subpopulation were not provided [85].

No study designed strictly to assess the effect of chemotherapy among patients with LNM after RP has been conducted thus far. In the STAMPEDE trial, the subset of patients with PCa and LNM (cN1 without previous local treatment or RT) in the subgroup treated with standard of care and docetaxel (*n* = 298) had favorable survival in comparison to patients with LNM treated only with standard of care (*n* = 296) (HR: 0.85; 95% CI 0.68–1.07) [86].

GETUG-12 is a randomized phase III clinical trial (NCT00055731) examining the role of first-line chemotherapy docetaxel and estramustine with ADT in the treatment of patients with high-risk localized PCa. The group of patients with LNM management with ADT plus docetaxel and estramustine (*n* = 59) had a beneficial impact on relapse-free survival at 8 years in comparison to the group treated with ADT only (*n* = 60) (HR 0.66, 95% CI 0.43–1.01; *p* = 0.017) [87].

On the other hand, the recent results from the randomized multinational phase III clinical trial of the Scandinavian Prostate Group 12 investigated the possible role of docetaxel in the treatment of patients with high-risk PCa (pT2 margin positive or pT3a Gleason score ≥ 4 + 3, pT3b, or lymph node positive disease Gleason score ≥ 3 + 4) after RP. In total, 459 patients were enrolled in the study and assigned to one of two groups: the first, those treated only with docetaxel (*n* = 230, 219 (95%) received at least one dose of docetaxel, 182 (79%) received all six cycles per protocol), and the second observation (*n* = 229). The statistical analysis did not reveal a favorable effect of the use of docetaxel after RP. Moreover, the same observation was made in the subgroup of patients with LNM (pN1) [88]. At the moment, there is one interventional, randomized phase II/III trial with open enrollment examining the impact of ADT and EBRT compared with or without docetaxel among patients after RP. The estimated primary outcomes will be revealed in 2026 [89]. A phase II clinical trial investigating the efficacy and safety of enzelatumid and standard care (ADT and RT) included eight patients with LNM after RP (eligible if fewer than three positive lymph nodes were dissected during RP and no lymph nodes > 2  cm were shown on screening imaging). Patients with LNM in comparison had poorer outcomes, with a 2-year progression-free survival rate of 25% (95% CI: 3.7, 55.8) with the historical control rate of 51% (95% CI: 33, 67) in a similar population of men with high-risk biochemically recurrent PCa [90]. One arm of the abovementioned STAMPEDE randomized controlled trial investigated the addition of abiraterone acetate and prednisolone to standard management (ADT and RT—depending on cofactors) in the treatment of non-metastatic PCa. In a population with local LNM, the use of abiraterone acetate, prednisolone, and ADT enhances failure-free (HR 0.26, 95% CI 0.17–0.40) and metastasis-free survival (HR 0.47 95% CI 0.29–0.78) [91].

## 5. Current Guidelines

The current guidelines differ from one another significantly, and they recommend a variety of management options for pN+ patients. The EAU identifies three major treatment possibilities: observation, early adjuvant HT, and ART with ADT. Observation is the strategy recommended as optimal for patients with 1–2 positive lymph nodes. Two remaining therapies are beneficial but require certain conditions to be fulfilled to use as a most advantageous treatment option. Early adjuvant HT significantly improves both CSS and OS, although available data on this subject include mostly patients with high-volume nodal burden and multiple adverse tumor characteristics. Adjuvant RT with ADT, on the other hand, is beneficial mostly for patients with PCa features such as <3 LNs, ISUP grade 2–5, pT3–4 or R1, or 3–4 positive nodes [73,74].

The NCCN indicates two groups of patients associated with LNM. The first one includes patients with high- or very-high-risk PCa with nodal metastasis discovered after RP. A patient with a high-risk disease is defined as having one of the following: T3a or GG 4–5 or PSA > 20 ng/mL; on the other hand, the very-high-risk group has the characteristics: T3b–T4 or primary Gleason pattern 5 or >4 biopsy cores with GG 4–5. For these patients the NCCN recommends only three options: ADT with or without EBRT or observation if postoperative PSA is undetectable. The second group includes patients with regional cancer that is described as N+ PCa of any size (any T); the nodal metastasis can be found during RP, PLND, or other tests. In these patients, several strategies are proposed on the basis of life expectancy. If life expectancy is 5 years or less and a patient has no symptoms, the guidelines suggest considering observation or ADT. Observation may be followed by palliative ADT if symptoms start to occur. However, if life expectancy is more than 5 years or a patient has symptoms, then five different strategies are mentioned: EBRT + ADT, EBRT + ADT + abiraterone, EBRT + ADT + fine-particle abiraterone, ADT + abiraterone, and ADT + fine-particle abiraterone. EBRT + ADT is considered a preferred solution [76].

The ESMO guidelines point out that ART with ADT shows good results for men with two positive lymph nodes associated with pT3b or pT4 and/or positive surgical margins when compared to RT alone [75].

NICE’s guidelines consider PCa with LNI as locally advanced PCa and suggest considering pelvic radiotherapy, but this guidance applies to patients before RP. Immediate post-operative radiotherapy, adjuvant hormonal therapy, and high-intensity focused ultrasound and cryotherapy are not recommended, and thus the most efficient management remains unclear [92]. Table 1 constitutes a detailed overview of the different management strategies indications in N+ patients provided by the most popular guidelines.

In summary, the unification of guidelines for approaches to the management of pN+ patients with PCa is strongly anticipated. The lack of clarity of the indications for this group of patients is the reason we did not describe the role of Canadian or American guidelines, which do not differentiate LN-positive patients as a separate group [93,94]. However, the guidelines described above fail to provide urologists with the most optimal management solutions. Future guidelines should thoroughly describe the management options while considering the heterogeneity of pN+ patients.

## 6. Conclusions

The optimal management of patients with lymph node metastases after RP is still unclear. These patients constitute a heterogeneous group and need substratification to choose the best therapeutic method, which will allow for the optimization of treatment results. The quality of current evidence is low, and most of the available studies are retrospective. However, new, ongoing clinical trials with existing and new therapeutic approaches are promising, and the results may help to systematize the guidelines.

## Figures and Tables

**Figure 1 cancers-14-02326-f001:**
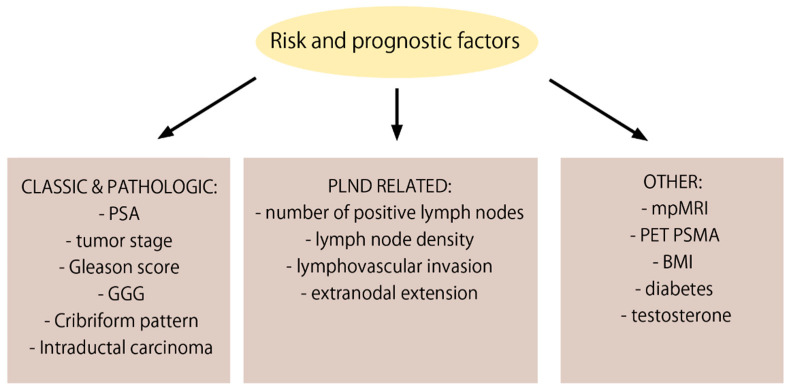
Risk and prognostic factors influencing outcomes in pN+ PCa patients after RP.

**Figure 2 cancers-14-02326-f002:**
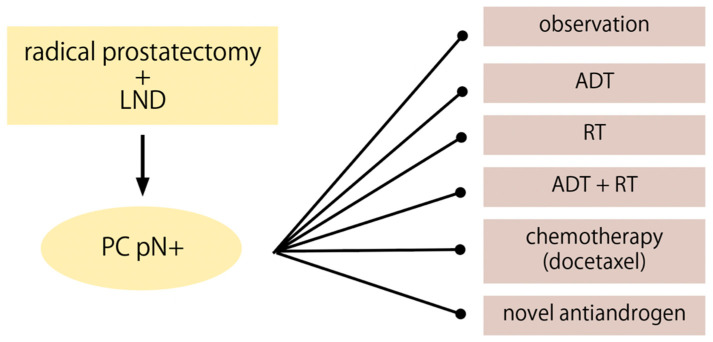
Therapeutic strategies in the management of patients with prostate cancer and lymph node metastases after radical prostatectomy (pN+).

**Table 1 cancers-14-02326-t001:** Overview of N+ patient management strategies according to guidelines provided by the EAU, the NCCN, the ESMO, and the NICE.

Guidelines	Specific Indications and Patient Characteristics	Management Strategies
EAU	- 1–2 positive LNs.	- Observation
	- Patients with high-volume nodal burden and multiple adverse tumor characteristics.	- Early adjuvant HT
	- <3 LNs, ISUP grade 2–5, pT3–4 or R1, or 3–4 positive nodes.	- ART with ADT
NCCN	- Undetectable postoperative PSA in patients with high- or very-high-risk pN+ PCa.	- Observation
	- Detectable postoperative PSA in patients with high or very-high-risk pN+ PCa.	- ADT with/without EBRT
	- Any N+ if life expectancy is 5 years or less and there are no symptoms.	- Observation (may be followed by palliative ADT if symptoms occur) or ADT
	- Any N+ if life expectancy is more than 5 years or a patient has symptoms.	- EBRT + ADT - EBRT + ADT + abiraterone - EBRT + ADT + fine-particle abiraterone - ADT + abiraterone - ADT + fine-particle abiraterone
ESMO	- Two positive LNs in patients with pT3b or pT4 and/or positive surgical margins.	- ART + ADT (guidelines indicate better results compared to RT alone)
NICE	- Nodal involvement in patients before RP.	- Pelvic RT

EAU: The European Association of Urology; NCCN: The National Comprehensive Cancer Network; ESMO: The European Society for Medical Oncology; NICE: National Institute for Health and Care Excellence; LN: lymph node; ISUP: International Society of Urological Pathology; PSA: prostate-specific antigen; PCa: prostate cancer; RP: radical prostatectomy; HT: hormone therapy; ART: adjuvant radiotherapy; ADT: androgen deprivation therapy; EBRT: external beam radiation therapy; RT: radiotherapy.

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
