# Peer review of "Patients with Positive Lymph Nodes after Radical Prostatectomy and Pelvic Lymphadenectomy—Do We Know the Proper Way of Management?"

_cancers, 2022, doi:10.3390/cancers14092326_

Round 1
Reviewer 1 Report
Dear Authors,
I read with interest your manuscript entitled: “Patients with positive lymph nodes after radical prostatectomy and pelvic lymphadenectomy – do we know the proper way of management?”.
The aim of the narrative review was to report the principal risk factors and therapies for PCa with LNI. Overall, the paper is interesting and well written.
Some major issues have to be corrected/discussed:
- The second chapter (risk or prognostic factors) is not in line with the title of your review: “There are many known risk factors associated with presence of lymph node metastses and worse prognosis in PCa patients”. Why you analysed risk factors associated with worse prognosis in PCa patients in a review based on patients with positive LN???
If you want to talk about risk factors, you have to focus their impact just on the risk of LNI: paragraphs 2.2, 2.3, 2.4, 2.5, 2.6, 2.7, 2.10, 2.11 and 2.12 have to be changed accordingly.
Moreover, paragraph 2.13 has to be removed (actually, no genomic or proteomic tests are used to establish the risk for LNI).
- Is there a difference in the management of a N+ patient depending on the extension of the lymphadenectomy (eg, standard vs extended vs super-extended)? Please discuss the topic in a dedicated paragraph
- Paragraph 3.5 is not based on N+ patients, please remove it
- What is the role of PET-PSMA in management of N+ patients?
- References:
- Chapter 2: there are several studies showing other features associated with lymph node invasion:
- Porcaro AB, Tafuri A, Sebben M, Processali T, Pirozzi M, Amigoni N, Rizzetto R, Shakir A, Cacciamani GE, Brunelli M, Siracusano S, Cerruto MA, Artibani W. Body Mass Index and prostatic-specific antigen are predictors of prostate cancer metastases in patients undergoing robot-assisted radical prostatectomy and extended pelvic lymph node dissection. Minerva Urol Nefrol. 2019 Oct;71(5):516-523. doi: 10.23736/S0393-2249.19.03401-5. Epub 2019 Jun 21. PMID: 31241272.
- Paragraph 2.12 mpMRI: re-write the paragraph focusing on the actual role of mpMRI in predicting the LNI prior to surgery:
- A Novel Nomogram to Identify Candidates for Extended Pelvic Lymph Node Dissection Among Patients with Clinically Localized Prostate Cancer Diagnosed with Magnetic Resonance Imaging-targeted and Systematic Biopsies. Gandaglia G, Ploussard G, Valerio M, Mattei A, Fiori C, Fossati N, Stabile A, Beauval JB, Malavaud B, Roumiguié M, Robesti D, Dell'Oglio P, Moschini M, Zamboni S, Rakauskas A, De Cobelli F, Porpiglia F, Montorsi F, Briganti A. Eur Urol. 2019 Mar;75(3):506-514. doi: 10.1016/j.eururo.2018.10.012. Epub 2018 Oct 17. PMID: 30342844
- Indication to pelvic lymph nodes dissection for prostate cancer: the role of multiparametric magnetic resonance imaging when the risk of lymph nodesinvasion according to Briganti updated nomogram is <5. Porpiglia F, Manfredi M, Mele F, Bertolo R, Bollito E, Gned D, De Pascale A, Russo F, Passera R, Fiori C, De Luca S.Prostate Cancer Prostatic Dis. 2018 Apr;21(1):85-91. doi: 10.1038/s41391-017-0026-5. Epub 2018 Feb 22.PMID: 29472629
Minor points
- Review the English grammar (some mistakes along the text)
- Try to add a table with a summary of the indications reported by different guidelines
Author Response
Response letter to the Reviewer #1 Report
We thank the Reviewer for encouraging feedback and appreciate the insightful comments and suggestions.
Below, we provide a point-by-point response to each of the reviewer’s comments.
All changes in the manuscript were highlighted in yellow for clarity.
We hope that the introduced revisions significantly improve the quality of this review and qualify it for further editorial stages.
Sincerely,
Authors
Major aspects:
- The second chapter (risk or prognostic factors) is not in line with the title of your review: “There are many known risk factors associated with presence of lymph node metastses and worse prognosis in PCa patients”. Why you analysed risk factors associated with worse prognosis in PCa patients in a review based on patients with positive LN???
If you want to talk about risk factors, you have to focus their impact just on the risk of LNI: paragraphs 2.2, 2.3, 2.4, 2.5, 2.6, 2.7, 2.10, 2.11 and 2.12 have to be changed accordingly.
Moreover, paragraph 2.13 has to be removed (actually, no genomic or proteomic tests are used to establish the risk for LNI).
Response: Thank you for this valuable suggestion. This paragraph was thoroughly rebuilt. We checked the available literature again and focused on the description of variables related to the presence of LNI. We also considered other factors in this section. Additionally, we have also rewritten the paragraph on mpMRI as well, taking into account the literature proposed by you, which we are grateful for.
- Is there a difference in the management of a N+ patient depending on the extension of the lymphadenectomy (eg, standard vs extended vs super-extended)? Please discuss the topic in a dedicated paragraph
Response: Thank you very much for this suggestion. We created a paragraph on impact of PLND templates – we briefly discussed its aims, different patterns and clinical outcomes, while addressing the issue of potential “overdiagnosis” of patients, referring to the Will-Rogers phenomenon.
- Include a Paragraph 3.5 is not based on N+ patients, please remove it.
Response: Thank you very much for this valuable comment. According to your suggestion, we have removed this paragraph, as it wasn’t most crucial part of the presented issue.
- What is the role of PET-PSMA in management of N+ patients?
Response: Thank you very much for this question and important suggestion. We have added another subsection in which we evaluated PSMA PET role in the management of lymph node positive patients and described its diagnostic efficiency briefly.
- References:
Chapter 2: there are several studies showing other features associated with lymph node invasion:
Porcaro AB, Tafuri A, Sebben M, Processali T, Pirozzi M, Amigoni N, Rizzetto R, Shakir A, Cacciamani GE, Brunelli M, Siracusano S, Cerruto MA, Artibani W. Body Mass Index and prostatic-specific antigen are predictors of prostate cancer metastases in patients undergoing robot-assisted radical prostatectomy and extended pelvic lymph node dissection. Minerva Urol Nefrol. 2019 Oct;71(5):516-523. doi: 10.23736/S0393-2249.19.03401-5. Epub 2019 Jun 21. PMID: 31241272.
Response: Thank you for this important advice. As mentioned above paragraph 2 was reconstructed and new features associated with LNI were described.
Paragraph 2.12 mpMRI: re-write the paragraph focusing on the actual role of mpMRI in predicting the LNI prior to surgery:
A Novel Nomogram to Identify Candidates for Extended Pelvic Lymph Node Dissection Among Patients with Clinically Localized Prostate Cancer Diagnosed with Magnetic Resonance Imaging-targeted and Systematic Biopsies. Gandaglia G, Ploussard G, Valerio M, Mattei A, Fiori C, Fossati N, Stabile A, Beauval JB, Malavaud B, Roumiguié M, Robesti D, Dell'Oglio P, Moschini M, Zamboni S, Rakauskas A, De Cobelli F, Porpiglia F, Montorsi F, Briganti A. Eur Urol. 2019 Mar;75(3):506-514. doi: 10.1016/j.eururo.2018.10.012. Epub 2018 Oct 17. PMID: 30342844
Indication to pelvic lymph nodes dissection for prostate cancer: the role of multiparametric magnetic resonance imaging when the risk of lymph nodesinvasion according to Briganti updated nomogram is <5. Porpiglia F, Manfredi M, Mele F, Bertolo R, Bollito E, Gned D, De Pascale A, Russo F, Passera R, Fiori C, De Luca S.Prostate Cancer Prostatic Dis. 2018 Apr;21(1):85-91. doi: 10.1038/s41391-017-0026-5. Epub 2018 Feb 22.PMID: 29472629
Response: Both references are added to revised paragraph 2.12 mpMRI (as mentioned above)
Minor aspects:
- Review the English grammar (some mistakes along the text)
Response: Article has undergone English language editing. The text has been checked for correct use of grammar and common technical terms and edited to a level suitable for reporting research in a scholarly journal.
- Try to add a table with a summary of the indications reported by different guidelines
Response: Thank you very much for pointing it out, we have added the table to the manuscript. Indeed, a table summarizing the most important guidelines add a great value to the article and should ease comprehension of discussed issues.

Reviewer 2 Report
The Authors present a review on a hot topic in PCa treatment. The paper is well written and all the aspects of the management of pN+ patients are highlighted
Author Response
Response letter to the Reviewer #2 Report
We thank the Reviewer for encouraging feedback and appreciate the insightful comments and suggestions.
Sincerely,
Authors

Reviewer 3 Report
Thank you for your submission. This is an interesting review article about clinical management of patients post-surgery, with many solid references.
There is minimal to no mention of recent PSMA related diagnostic and therapeutic options. It might be helpful to add some paragraph and references.
Here are some minor issues:
- Line 24: a heated issue, should be controversial issue or hotly debated issue
- Line 38: 1.4 mln, need to spell out
- May need to have a native English speaker to edit more of syntax
Author Response
Response letter to the Reviewer #3 Report
We thank the Reviewer for encouraging feedback and appreciate the insightful comments and suggestions.
Below, we provide a point-by-point response to each of the reviewer’s comments.
All changes in the manuscript were highlighted in yellow for clarity.
We hope that the introduced revisions significantly improve the quality of this review and qualify it for further editorial stages.
Sincerely,
Authors
Major issue:
- There is minimal to no mention of recent PSMA related diagnostic and therapeutic options. It might be helpful to add some paragraph and references.
Response: Thank you very much for this important suggestion. We have added another subsection in which we evaluated PSMA PET role in the management of lymph node positive patients and described its diagnostic efficiency.
Minor issues:
- Line 24: a heated issue, should be controversial issue or hotly debated issue
Response: Thank you for pointing out this stylistic error. We‘ve corrected this sentence.
- Line 38: 1.4 mln, need to spell out
Response: Corrected in text.
- May need to have a native English speaker to edit more of syntax
Response: Article has undergone English language editing. The text has been checked for correct use of grammar and common technical terms and edited to a level suitable for reporting research in a scholarly journal.

Round 2
Reviewer 1 Report
Authors modified the manuscript in line with reviewer's suggestions.